# Prognosis Discussion and Referral to Community Palliative Care Services in Patients with Advanced Pancreatic Cancer Treated in a Tertiary Cancer Centre

**DOI:** 10.3390/healthcare11202802

**Published:** 2023-10-23

**Authors:** Sarah Clelland, Christina L. Nuttall, Helen E. Stott, Joseph Cope, Natalie L. Barratt, Kelly Farrell, Manyi V. Eyong, Jack P. Gleeson, Angela Lamarca, Richard A. Hubner, Juan W. Valle, Mairéad G. McNamara

**Affiliations:** 1The Christie NHS Foundation Trust, Manchester M20 4BX, UK; 2Division of Cancer Sciences, University of Manchester, Manchester M20 4BX, UK

**Keywords:** prognosis, pancreatic cancer, palliative care, advance care planning, nurse practitioner, clinical nurse specialist

## Abstract

Advanced pancreatic cancer is associated with a poor prognosis, often less than 1 year. Honest prognosis discussions guide early community palliative care services input, facilitating timely advance care planning and improving quality of life. The aims were to assess if patients were offered prognosis discussions and community palliative care services referral. A retrospective analysis of consecutive case-notes of new advanced pancreatic cancer patients was conducted. Chi-squared test assessed the association with prognosis discussion and community palliative care services referral. In total, 365 cases (60%) had a documented prognosis discussion at any time-point in the treatment pathway; 54.4% during the first appointment. The frequency of prognosis discussion was greater with nurse clinician review at first appointment (*p* < 0.001). In total, 171 patients (28.1%) were known to community palliative care services at the first appointment. Of those not known, 171 (39.1%) and 143 (32.7%) were referred at this initial time-point or later, respectively. There was a significant association between the referral to community palliative care services at first appointment and the reviewing professional (this was greatest for nurse clinicians (frequency 65.2%)) (*p* < 0.001), and also if reviewed by clinical nurse specialist at first visit or not (47.8% vs. 35.6%) (*p* < 0.01). Prognosis discussions were documented in approximately two-thirds of cases, highlighting missed opportunities. Prognosis discussion was associated with clinician review and was most frequent for nurse clinician, as was referral to community palliative care services. Clinical nurse specialist review increased referral to community palliative care services if seen at the initial visit. Multi-disciplinary review, specifically nursing, therefore, during the first consultation is imperative and additive. It should be considered best practice to offer and negotiate the content and timing of prognosis discussions with cancer patients, and revisit this offer throughout their treatment pathway. Greater attention to prognosis discussion documentation is recommended.

## 1. Introduction

Approximately 80% of patients with pancreatic cancer have advanced disease at diagnosis [1]. This is associated with a poor prognosis: the median survival for locally advanced disease is 6–11 months and can reduce to 2–12 months for those with metastatic disease, with or without systemic treatment [2].

Treatment options for patients with pancreatic cancer are limited. If surgery is an option, this achieves a 5-year survival rate of approximately 10% [3]. The first-line chemotherapy treatment option for patients with unresectable locally advanced or metastatic pancreatic cancer who have poorer performance status has been Gemcitabine since 1997 [4]. Combination chemotherapy has modestly improved survival to approximately 9 months with Gemcitabine and Nab-paclitaxel [5] or 12 months for patients fit enough to receive FOLFIRINOX [6].

Patients with advanced pancreatic cancer are often, therefore, in their last year of life at the time of diagnosis but may also, depending on their performance status, be offered palliative systemic anti-cancer treatment. 

It is noted that timely specialist palliative care input in patients with advanced cancer has been shown to improve outcomes in care planning [7] and that discussions regarding end of life care are associated with less aggressive medical care, impacting quality of life [8]. Given the low survival rates in patients with advanced pancreatic cancer, honest prognosis discussions are crucial in helping to guide referral to community palliative care services, with a view to facilitating timely advance care planning.

It is not known how often prognosis discussions are offered and recorded by oncologists and the primary aim of this study, therefore, was to determine the frequency of prognosis discussions with patients with advanced pancreatic cancer in a tertiary cancer centre either during their first clinic appointment, or at any time point following this. The secondary objective was to assess the frequency of referrals to community palliative care services in this patient cohort.

## 2. Materials and Methods

This study included patients age ≥ 18 years with a tissue or radiological diagnosis of pancreatic cancer that was locally advanced or metastatic, and not amenable to curative surgical resection. All the patients had been referred to the hepato–pancreato–biliary oncology disease group at The Christie NHS Foundation Trust (“The Christie”) between December 2012 and February 2019, for consideration of palliative systemic anti-cancer treatment, or for a second opinion. These dates were selected based on patient information being available electronically via a Cancer Oncology Group form. This form is completed by the clinician (doctor or nurse) who undertakes the initial new patient consultation at the first visit. There is no formal process in place for triaging which patients are seen by which type of clinician. This is at random and may be a consultant, specialist registrar (doctor in a speciality training programme working in the UK), clinical fellow or nurse clinician. Nurse clinicians, also known as advanced nurse practitioners, are nurses who have completed a Master’s Level education programme in clinical practice. They are capable of working independently in the assessment, diagnosis and treatment of patients and are independent non-medical prescribers. This is in contrast to ‘registered nurses’ who have not undertaken this additional education. Registered nurses deliver patient care as specified by clinicians and do not work autonomously in this regard.

Data were collected by analysing patient electronic medical notes retrospectively, using the Clinical Web Portal, which is hospital specific. The Cancer Oncology Group form, which is completed on the patient’s first visit to The Christie, provided information on the patient demographics, diagnosis, cancer staging and Eastern Cooperative Oncology Group performance status (ECOG PS). Other data captured included if treatment commenced; if prognosis discussions were offered and the rationale for declining (if relevant); and if referred to community palliative care services. Community palliative care services provide holistic medical and nursing care, with a particular focus on quality of life, in a patient’s own home. Palliative care services offered at The Christie can only be accessed from within the hospital. This is therefore not suitable for most patients who will attend The Christie infrequently and may receive treatment and ongoing monitoring at a satellite site.

Additional information collected for this study included the details of the clinician who reviewed the patient at their first visit and whether the patient met the clinical nurse specialist. Clinical nurse specialists practice advanced-level nursing but have a different role to nurse clinicians. They act as keyworkers helping patients to manage symptoms, deal with practical issues and support with emotional wellbeing. They also help to coordinate care and treatment plans.

All data were stored in a secure database and patient identifiable details were anonymised. Data collection was cross-checked with original data from the Clinical Web Portal for accuracy.

A Chi-squared test was used to assess for statistical significance. 

Final analysis and results were then member checked by all authors on review of collated findings.

This project was approved by the Quality Improvement and Clinical Audit team at The Christie (reference 2690) and ethical approval was therefore not required.

## 3. Results

In total 703 case-notes were reviewed. Overall, 95 patients with resected disease were excluded and the eligible study cohort (*n* = 608) was made up of predominantly (93.9%) those with a histological diagnosis of advanced pancreas adenocarcinoma (Figure 1).

Just over half of the study cohort were male (54.1%) and the majority were ECOG PS 1 (53.8%). A small proportion had a performance status 3 and 4 (14.8%) (Table 1).

The majority of patients had metastatic disease (61.3%). One-third received best supportive care alone and there was a relatively even division between the treatment decisions for triplet chemotherapy, doublet chemotherapy and monotherapy (Table 1).

The majority of the Cancer Oncology Group forms were completed by a consultant at the first clinic appointment (50.5%) (Figure 2).

The ECOG PS distribution was similar for those patients seen at first clinic appointment by the nurse clinician when compared to those seen first by a doctor. The nurse clinician saw 40.7% patients who were for best supportive care, compared to 31.7% of those seen by a doctor (Table 2).

Prognosis discussions were documented at any time-point in the treatment pathway in 365 (60%) cases and for just over half (330, 54.4%) this was during their first new patient appointment. A reason for why a prognosis discussion did not take place at this time was documented in 112 cases (40.5%). The most common documented reason was that the patient did not wish to have this discussion (98, 35.4%). There was a statistically significant association between the frequency of prognosis discussions at the first clinic appointment and the clinician completing the Cancer Oncology Group form. This was greatest if the patient was seen by the nurse clinician (frequency 81.5%) (*p* < 0.001). 

Eight further patients (1.3%) had a prognosis discussion within the 30 days after their first clinic appointment and 28 (4.6%) had a prognosis discussion after 30 days.

Patients who had a treatment decision of best supportive care were most likely to be offered a prognosis discussion (66.5%) but this was not dramatically high compared to those who went on to have chemotherapy, as prognosis discussion rates ranged in this group between 50% and 60%.

Approximately 80% of patients were referred to community palliative care services, either prior to being seen in the cancer centre or at some time-point in their treatment pathway (Figure 3).

There was a significant association between referral to community palliative care services at the first clinic appointment and the clinician completing the Cancer Oncology Group form. This was greatest for the nurse clinician (frequency 65.2%) (*p* < 0.001). 

Approximately one-third (34.0%) were seen by a clinical nurse specialist at their first clinic appointment and there was also a significant association between this and referral to community palliative care services (frequency 47.8%) or not (frequency 35.6%) (*p* < 0.01). The majority of the patients seen by the clinical nurse specialists were ECOG PS 1 (56.1%) and there was an even distribution of treatment decisions for this cohort (Table 2).

## 4. Discussion

### 4.1. Main Findings

Approximately two-thirds of patients were documented to have been offered a prognosis discussion whilst under the care of the hepato–pancreato–biliary disease medical oncologists. The most common documented reason why a prognosis discussion did not take place was that the patient did not wish to have this conversation. The frequency of prognosis discussions was best for the nurse clinician, as was referral to community palliative care services. Approximately 80% of the study cohort were either already known to community palliative care services or were documented as referred during their treatment pathway. Referral to community palliative care services was increased by being seen by the clinical nurse specialist at the patient’s first clinic review.

### 4.2. What This Study Adds

The demographic data show that the majority of patients being referred to medical oncology for consideration of active cancer treatment have an ECOG PS of 0 to 2. This represents the sub-group of patients who are likely to be fit enough to tolerate systemic treatment and suggests that most referrals are appropriate. Ninety patients had an ECOG PS of 3 or 4. However, 200 patients had a treatment decision of best supportive care after their initial new patient review. This indicates that factors other than ECOG PS play a role in deciding on treatment intent, likely including biochemical and haematological parameters, co-morbidities and potentially patient wishes. 

This study showed that being seen by the nurse clinician at the first clinic review was significant for being offered a prognosis discussion and for referral to community palliative care services. The ECOG PS of the patient cohort seen by the nurse clinician was similar to that of those seen by doctors. However, the nurse clinician reviewed a 9% higher percentage of patients who were for best supportive care. This small difference in patient cohort may partly explain our findings; however, nurse clinicians are recognised for improving clinical outcomes and patient satisfaction [9,10] and the difference is not thought large enough to be the only factor. 

Additionally, being seen by the clinical nurse specialist was associated with increased referral to community palliative care services. Most patients seen by the clinical nurse specialist were ECOG PS 1 and the majority had a treatment decision of systemic anti-cancer treatment. This highlights their valuable role as part of the multi-disciplinary team in improving care for patients with cancer, in line with established evidence [11]. 

Both nurse clinicians and clinical nurse specialists are considered advanced nursing roles. One study examining nurses working in similar roles in primary care found that they were able to spend longer with patients during consultations and provided patients with more information about their illness [12]. Another study reported that patients find these advanced-level nurses easier to talk to, leading to higher levels of patient satisfaction [13]. Our work furthers existing evidence, showing that multi-disciplinary input during first clinic review, particularly from advanced-level nursing roles, improves the offer of both prognosis discussions and referral to community palliative care services. One possible explanation for this could be due to spending more time with the patient and their family (with some physicians possibly being more time-constrained due to the additional needs associated with providing consultation to all other healthcare workers in clinic in addition to seeing the patients), allowing for a more trusting relationship to be developed, which lends itself to having potentially difficult conversations about prognosis.

We therefore make the case for the diversification of healthcare professionals within oncology teams and suggest that this may improve outcomes relating to prognosis discussions and referral to community palliative care services. Future work to improve these outcomes further could focus on triaging which patients are seen by the nurse clinician and clinical nurse specialist.

### 4.3. Strengths/Limitations

A literature review was performed across three databases (CINAHL, EMBASE, MEDLINE) in October 2021 with the terms prognosis discussions, conversations, consultations and communications for solid organ tumours. There is a large body of evidence examining the role of palliative care consults on patient outcomes but less work focussing specifically on prognosis discussion. 

One Australian survey of patients with metastatic disease found that 81% wished to discuss expected survival at some time-point in their treatment pathway (59% when first diagnosed) [14]. In our study, 60% had documentation to confirm they had been offered a prognosis discussion but only 5.6% had a documented prognosis discussion at a time-point after their first clinic review. In our tertiary cancer centre, there is, therefore, likely to be a proportion of patients who would wish to discuss prognosis after their first clinic review and who are not currently being offered this opportunity. Given this, we recommend that clinicians revisit these discussions at subsequent consultations. This is important as a patient’s prognosis may change during their disease trajectory and also because a patient’s wishes regarding the information they want to receive may change.

This study relied on the accurate documentation of consultation discussions and, therefore, it is possible that a higher proportion of patients were offered a prognosis discussion, but that this was not recorded in the medical notes. Amendments to the mandatory data entries in the Cancer Oncology Group form may help to improve this.

Current evidence suggests that early conversations focussing on “goals of care”—including prognosis discussions—are associated with improved quality of life and family experiences; and less exposure to aggressive medical care near the end of life, with a reduction in costs [8,15]. Prognosis discussions support patients with medical decision making and end of life care planning and having these conversations early is likely to improve the quality of this planning for the patient and their family [16,17]. This is of particular importance in advanced pancreas cancer due to its poor prognosis. One cluster randomised controlled trial examining a model of training clinicians in how to conduct such conversations found a statistically significant reduction in anxiety and depression in patients who participated [18,19]. However, a study in 2023 by Pihlak et al., found discrepancies between patients’ and clinicians’ views regarding aims and priorities for treatment [20]. Therefore, whilst it would seem best practice to offer prognosis discussions and negotiate the timing and content of these with patients, surveying patients to gather their views regarding this is essential. This represents an area for future work, particularly given that we found the most common reason a prognosis discussion did not take place was because the patient did not wish to have this conversation.

This study does not examine the potential barriers that clinicians perceive in relation to offering prognosis discussions. Factors cited in the literature include uncertainty about estimating survival and also worries about causing patient distress or harm [21]. However, a study by Fenton et al. in 2017 demonstrated that prognostic discussion was not intrinsically harmful and may actually strengthen the doctor–patient relationship [22]. Our findings may expose that doctors, as a specific cohort, are less good at offering these discussions and surveying clinicians working within tertiary cancer centres to better understand the reasons why prognosis discussions are, or are not, being offered may help to target interventions for improvement.

Almost 80% of patients were referred to community palliative care services, either prior to being seen in the medical oncology department, or after. It is not possible to know whether the referral resulted in the patient definitely being seen by a palliative care specialist. However, given the evidence that specialist palliative care input is associated with improved outcomes [8,15]—and that palliative care referral is recommended by the American Society of Clinical Oncology for all patients with advanced cancer [23]—this is encouraging. A study by Lees [24] looking specifically at the impact of palliative care input for patients with unresectable pancreatic cancer found that regardless of when this input occurred, it was associated with less aggressive care at the end of life, suggesting that community palliative care service referral does not necessarily need to be discussed at first clinic review. Furthermore, a greater proportion of our cohort were referred to community palliative care services than had prognosis discussions, indicating that patients still consent to receiving specialist palliative care input despite not necessarily being aware of their prognosis. It would be helpful to understand the reasons why 20% were not referred to community palliative care services, as this could be patient choice (perceived lack of need, reluctance to engage with end of life care) or that they were already receiving specialist palliative care input in another form elsewhere. Again, this indicates an area for future research.

Finally, it is important to acknowledge the limited resources of specialist palliative care [25]. With the increase in cancer treatments resulting in a growing population of patients who are treatable, but not curable, it may not be feasible for every patient with metastatic or advanced cancer to be seen by community palliative care services. Consideration of alternative models of working to address this have been examined and, given that a proportion of the cohort included in this study went on to receive systemic anti-cancer treatment, a supportive care model involving the integration of multiple clinical specialities may be appropriate for these patients in the future [26].

## 5. Conclusions

Given the poor survival for patients with advanced pancreatic cancer, the offer of early honest prognosis discussions can help improve quality of life, but this does not always occur. This was concluded following the data collection for this study, where a prognosis discussion was documented to have been offered in approximately two-thirds of cases, acknowledging that all discussions may not have been recorded. Greater attention to prognosis discussion documentation is therefore recommended. 

Prognosis discussion was associated with the person completing the Cancer Oncology Group form, as was referral to community palliative care services, both of which were most frequent for nurse clinicians. Being seen by the clinical nurse specialist at initial visit also resulted in increased referral to community palliative care services. Multi-disciplinary review, specifically nursing, therefore, during the first consultation is imperative and additive. 

It should be considered best practice to offer and negotiate the content and timing of prognosis discussions but future surveys to assess whether patients would favour a prognosis discussion in the initial consultation could be considered. Revisiting the prognosis discussion during the patient pathway is recommended. It would also be interesting to assess the viewpoint of the clinicians in relation to discussion of prognosis in clinic.

## Figures and Tables

**Figure 1 healthcare-11-02802-f001:**
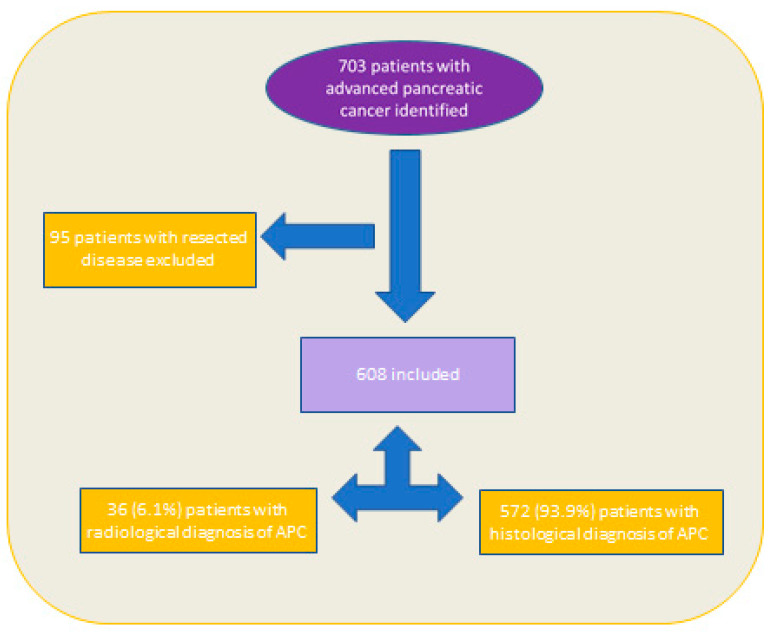
Schema of inclusion for patients with advanced pancreatic cancer. APC: advanced pancreatic cancer.

**Figure 2 healthcare-11-02802-f002:**
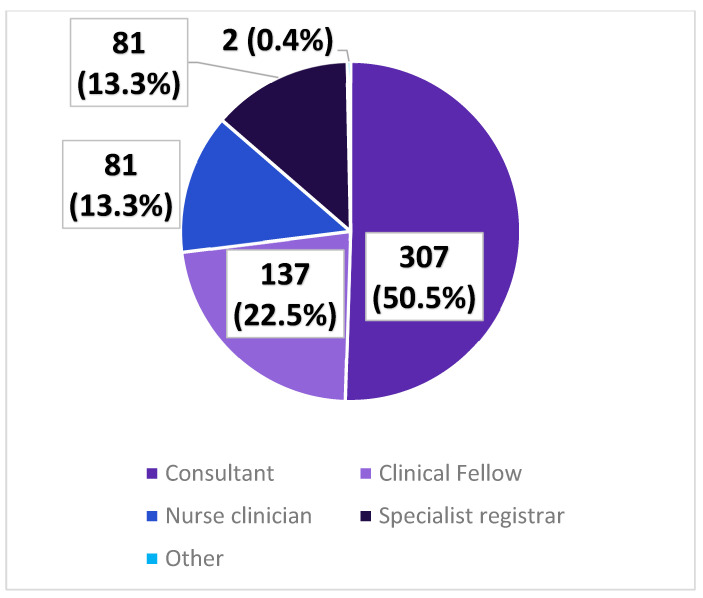
The distribution of clinicians completing the Cancer Oncology Group form at first clinic appointment.

**Figure 3 healthcare-11-02802-f003:**
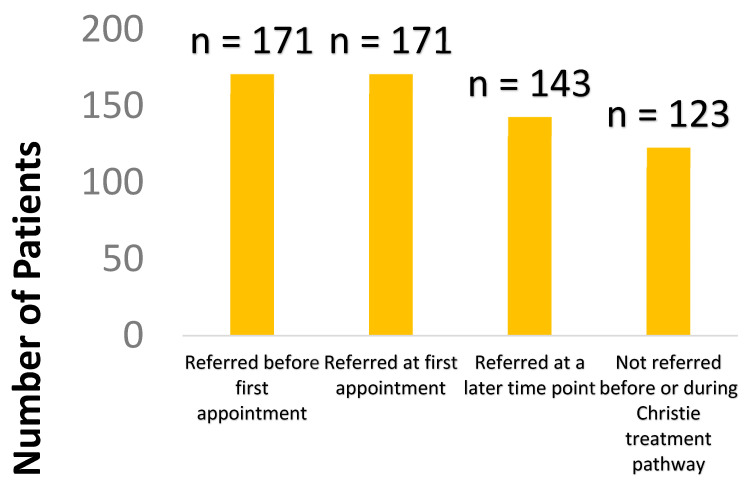
When patients were referred to community palliative care services in relation to their first appointment.

**Table 1 healthcare-11-02802-t001:** Demographics of included patients with advanced pancreatic cancer.

Variable	Category	*n* (%)
Age	Median (range)	68 years (25–94)
Gender	Male	329 (54.1)
	Female	279 (45.9)
Stage	Locally advanced	229 (37.7)
	Metastatic	373 (61.3)
	Recurrent	3 (0.5)
	Unknown	3 (0.5)
Eastern Cooperative Oncology Group Performance Status (ECOG PS)	0	45 (7.4)
	1	327 (53.8)
	2	146 (24.0)
	3	85 (14.0)
	4	5 (0.8)
Treatment Decision	Triplet chemotherapy	117 (19.2)
	Doublet chemotherapy	141 (23.2)
	Monotherapy	144 (23.7)
	Best supportive care	200 (32.9)
	Other	6 (1.0)

Other: Those categorised as other reflect patients who attended for a second opinion and received their treatment elsewhere; and/or where this information was unknown.

**Table 2 healthcare-11-02802-t002:** Demographics of those patients seen at first clinic appointment by nurse clinician, doctor and clinical nurse specialist.

Variable	Category	Nurse Clinician (*n* = 81)	Doctor (Consultant, Clinical Fellow, Specialist Registrar or Other) (*n* = 527)	Clinical Nurse Specialist (*n* = 187)
Eastern Cooperative Oncology Group performance status	0	7.4% (6)	7.4% (39)	7.5% (14)
	1	49.4% (40)	54.5% (287)	56.1% (105)
	2	25.9% (21)	23.7% (125)	27.8% (52)
	3	17.3% (14)	13.5% (71)	7.5% (14)
	4	0% (0)	0.9% (5)	1.1% (2)
Treatment Decision	Triplet chemotherapy	11.1% (9)	20.5% (108)	20.9% (39)
	Double chemotherapy	24.7% (20)	23.0% (121)	25.7% (48)
	Monotherapy	23.5% (19)	23.7% (125)	26.2% (49)
	Best supportive care	40.7% (33)	31.7% (167)	27.2% (51)
	Other	0% (0)	1.1% (6)	0% (0)

## Data Availability

The data that supports the findings of the study are stored in a secure drive in the hepato–pancreato–biliary disease group at The Christie NHS Foundation Trust. Data are not publicly available but can be obtained from the authors upon reasonable request.

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
