# Peer review of "Prognosis Discussion and Referral to Community Palliative Care Services in Patients with Advanced Pancreatic Cancer Treated in a Tertiary Cancer Centre"

_healthcare, 2023, doi:10.3390/healthcare11202802_

Round 1

Reviewer 1 Report

The title of the work: Prognosis discussion and referral to community palliative care 2 services in patients with advanced pancreatic cancer treated in a 3 tertiary cancer centre. The main aims of the authors were the assess the feasibility of the „prognosis discussion” and the frequency of the early access into palliative care in case of advanced pancreatic cancer patients in a solitary tertiary cancer centre. So, the general purposes, moreover the main messages of the work are clear, simply, the manuscript is well-written and informative. However, I missed some more detailed discussion about the factual role of the clinical nurse specialists, some explanation why their (and the clinicians) communication is more acceptable for the patients and I missed some thought about the significance of the early prognosis discussion (over the significance of the early enter to palliative care). Nevertheless, I consider the paper interesting, important and I recommend it to consider to publication.

One extra note: In Figure 3. the sum of the percentages is over 100%. Please clarify or explain it. 

Reviewer 2 Report

Thank you to the authors for submitting this paper on an important topic. 

In general, the introduction and discussion are well written.  However, it is difficult to draw conclusions about the impact of professional background on outcomes without more information about the process of patient assignment and potential influence of patient characteristics.  Also, it would be helpful to have clarity around terminology and more information about palliative care services.  Specific comments are below:

Methods

Define specialist registrar for non-UK readers.

How were patients assigned to physicians versus nurses for initial consultation?  Was there a triage process?

Are registered nurse, nurse clinician, advanced nurse practitioner and clinical nurse specialist the same?  In some jurisdictions, registered nurse, advanced nurse practitioner and clinical nurse specialist have different educational backgrounds and scopes of practice.

What do "community palliative care services" comprise?  Are any palliative care services available at the oncology facility?

Results

It would be helpful to analyze for differences in patient characteristics between patients seen by physicians versus nurses (e.g., ECOG).

Figure 2: Use consistent terminology for "Nurse clinician".

Figure 3: The percentages do not add up to 100.  Please clarify.  

Discussion

It is difficult to conclude that increased prognosis discussions and referrals to community palliative care services by nurses were the result of their professional background, without knowing whether their patients differed or not from those seen by physicians (e.g., did they have poorer performance status or more comorbidities and therefore were less likely to proceed to systemic therapy).  Please address.

Round 2

Reviewer 2 Report

Thank you for the revisions.  My comments have been addressed satisfactorily.